# Microbiome Changes after Type 2 Diabetes Treatment: A Systematic Review

**DOI:** 10.3390/medicina57101084

**Published:** 2021-10-11

**Authors:** Kajus Merkevičius, Ričardas Kundelis, Almantas Maleckas, Džilda Veličkienė

**Affiliations:** 1Faculty of Medicine, Vilnius University, LT-03101 Vilnius, Lithuania; ricardas.kundelis@gmail.com; 2Department of Surgery, Medical Academy, Lithuanian University of Health Sciences, LT-50009 Kaunas, Lithuania; almantas.maleckas@lsmu.lt; 3Department of Surgery, Institute of Clinical Sciences, Sahlgrenska Academy, University of Gothenburg, 405 30 Gothenburg, Sweden; 4Institute of Endocrinology, Lithuanian University of Health Sciences, LT-50009 Kaunas, Lithuania; Dzilda.Velickiene@lsmuni.lt

**Keywords:** type 2 diabetes, gut microbiome, bariatric surgery, probiotics, prebiotics, synbiotics

## Abstract

*Background and objectives*: Although the role of the gut microbiome in type 2 diabetes (T2D) pathophysiology is evident, current systematic reviews and meta-analyses analyzing T2D treatment mainly focus on metabolic outcomes. The objective of this study is to evaluate the microbiome and metabolic changes after different types of treatment in T2D patients. *Materials and Methods*: A systematic search of PubMed, Wiley online library, Science Direct, and Cochrane library electronic databases was performed. Randomized controlled clinical trials published in the last five years that included T2D subjects and evaluated the composition of the gut microbiome alongside metabolic outcomes before and after conventional or alternative glucose lowering therapy were selected. Microbiome changes were evaluated alongside metabolic outcomes in terms of bacteria taxonomic hierarchy, intestinal flora biodiversity, and applied intervention. *Results*: A total of 16 eligible studies involving 1301 participants were reviewed. Four trials investigated oral glucose-lowering treatment, three studies implemented bariatric surgery, and the rest analyzed probiotic, prebiotic, or synbiotic effects. The most common alterations were increased abundance of *Firmicutes* and *Proteobacteria* parallel to improved glycemic control. Bariatric surgery, especially Roux-en-Y gastric bypass, led to the highest variety of changed bacteria phyla. Lower diversity post-treatment was the most significant biodiversity result, which was present with improved glycemic control. *Conclusions*: Anti-diabetic treatment induced the growth of depleted bacteria. A gut microbiome similar to healthy individuals was achieved during some trials. Further research must explore the most effective strategies to promote beneficial bacteria, lower diversity, and eventually reach a non-T2D microbiome.

## 1. Introduction

Diabetes ranks among top ten leading causes of mortality worldwide, and by the year 2045 diabetes cases are expected to reach 700 million [1]. In the last decade, the role of the gut microbiome, the genome of all intestinal microorganisms, in the pathophysiology of type 2 diabetes (T2D) gained increasing attention from the scientific community. Animal models and studies involving humans resulted in significant differences in the microbiome of healthy and T2D subjects. Furthermore, it is reasonably believed that, among other mechanisms, microbial disorders lead to low-grade intestinal inflammation and insulin resistance, which are closely related to the cause of T2D [2,3,4].

During the last five years, several systematic reviews and meta-analyses emerged that evaluated the effectiveness of different approaches to treat T2D in humans by influencing intestinal microflora [5,6,7,8,9,10,11]. Studies reviewing dietary interventions have resulted in significant results regarding fasting blood glucose, hemoglobin A1c (HbA1c), or lower insulin resistance [5,6,7,8]. Nevertheless, only two systematic reviews evaluated changes of the gut microbiome after applied treatment [7,8]. Studies that reviewed metabolic and microbiome changes after bariatric surgery included animal trials or subjects without T2D [10,11]. Thus, a systematic review that would evaluate both glucose-lowering effects and changes in the gut microbiome after T2D treatment in human subjects is needed.

This systematic review aims to evaluate the microbiome and metabolic changes after different types of treatment in T2D patients. This study reviewed randomized, controlled clinical trials that included only human T2D subjects and evaluated the composition of the gut microbiome before and after treatment, which consisted of conventional glucose-lowering therapy; probiotic, prebiotic, or synbiotic supplementation; or bariatric surgery.

## 2. Materials and Methods

This systematic review was carried out in accordance with the preferred reporting item for systematic reviews and meta-analyses (PRISMA) guidelines (see Appendix A PRISMA checklist) [12,13]. Prior to first publication no protocol of this review was registered in any database.

### 2.1. Eligibility Criteria

Studies were included in the review if they met the criteria of study type, characteristics of participants, applied intervention, and outcome measurements.

Eligible study designs were randomized controlled trials (RCTs) that were conducted in a single-blind, double-blind, or unmasked manner. Both placebo (no treatment) and different types of intervention (bariatric surgery, anti-diabetic medication, prebiotics, or probiotics) used as comparator were included. Two-arm or multiple-arm study designs were applied. Only one animal study was included, as it had a parallel part involving human subjects and trial parts involving animals and humans did not interfere with one another.

Inclusion criteria: subjects with type 2 diabetes, aged 18 and older. The duration of diabetes, prior treatment status, other diseases, and level of HbA1c had no effect on eligibility of the studies. Regarding applied treatment, standard T2D treatment or novel alternatives that are hypothesized to improve glycemic control or lower insulin resistance were included. Alternatives eligible for consideration were oral probiotics, prebiotics, or synbiotics; bariatric surgery; and fecal matter transplantation. Trials that investigated the effects of depletion diets were excluded.

Finally, for a study to be eligible for further analysis in this systematic review, the outcomes had to consist of the evaluation of the gut microbiota’s composition and measurements representing glycemic control. Microbiome evaluation would be eligible if the abundance of bacteria was measured with or without diversity assessment. These measurements had to be reported from at least one fecal sample collected before and after the applied intervention. All storage settings, evaluation techniques, and testing equipment used to analyze these samples were permitted. Moreover, serum glucose level (either fasting or postprandial) or HbA1c had to be measured before and after the applied intervention. Other optional anthropometric and metabolic parameters would be included if they were recorded both before and after.

### 2.2. Search

Studies were identified in PubMed, Wiley online library, Science Direct, and Cochrane library electronic databases. The continuous search process was initiated in 2019/11 and completed in 2020/11. RCTs in any language that were published in the last five years were included. Search keywords and applied strings are listed in the Appendix A. Additional search through reference lists within studies fulfilling the eligibility criteria was performed. Duplicates were removed.

### 2.3. Study Selection

Authors independently reviewed the titles and abstracts of all the studies generated by the search to identify trials that met the inclusion criteria detailed above. Full-text articles were assessed when the abstract lacked sufficient information or whenever a disagreement was present on whether a trial should be kept for further analysis. Full-text assessment of the selected studies was performed by all authors. An independent reviewer (A.M.) evaluated all steps and decisions of the study selection process. The final decision on whether the trials in question should be retained in this systematic review was made by reaching a consensus.

### 2.4. Data Extraction

Key data were extracted from all selected trials. No pre-piloted form was used. Two authors (K.M., R.V.) extracted all data while a third author (D.V.) independently checked the accuracy of data entry. Any disagreements were solved by reaching a consensus. An independent reviewer was called upon when unanimous decision was absent. Data included general information, number of participants, type and duration of the applied intervention, follow-up period, outcomes reported, and other relevant findings. Original authors were contacted if data regarding microbiome changes were missing. The extracted data were summarized in tables that represent general characteristics of selected trials, changes in the composition of intestinal microbiome, and the outcomes of different interventions. All extracted data were represented as nominal data. No synthesis or statistical data analysis was performed.

### 2.5. Risk of Bias and Quality

To ascertain the validity of eligible randomized trials, the risk-of-bias assessment was performed using the revised Cochrane risk-of-bias tool (RoB2) [14]. A standardized questionnaire based on RoB2 was used to assess the quality of each included clinical trial. Studies were evaluated for the following items: bias arising from the randomization process, bias due to deviations from intended interventions, bias due to missing outcome data, bias in measurement of the outcome, and bias in selection of the reported results. The overall risk-of-bias judgement for a specific result was concluded based on questionnaire results of each bias domain in accordance with the RoB2 algorithm. The assessment was performed by three authors (K.M., R.K., D.V.) and checked by an independent reviewer. The same reviewer assessed the quality of this systematic review in accordance with study quality assessment tool provided by the National Heart, Lung, and Blood Institute (NHLBI) [15].

## 3. Results

A total of 1139 records were identified in the databases search, and an additional search in references of eligible studies pointed out 14 records. Figure 1 shows the PRISMA flow diagram that provides details of the inclusion process. After a full-text assessment, 16 randomized controlled trials with a total of 1301 participants were included in the systematic review. The characteristics of all selected trials and risk-of-bias evaluation results are summarized in Table 1. Out of 16 trials, 2 had low and 8 had high risk-of-bias, the rest had some concerns.

### 3.1. Oral Anti-Diabetic Treatment

Four RCTs of antidiabetic treatment fulfilled inclusion criteria in the study. The efficacy to alter intestinal microbiome and associated metabolic outcome of oral anti-diabetic medications was assessed by evaluating metformin, acarbose, and glipizide [16,17,18,19]. The latter was compared with acarbose in a trial by Gu et al. and resulted in distinctive differences (Table 2).

First, after 12 weeks glipizide produced only a decrease in serum glucose and HbA1c, while acarbose group additionally resulted in significantly lower body weight, BMI (*p* = 3.48 × 10–7), and HOMA-IR (*p* = 0.002) along with lower total cholesterol (*p* = 0.007) and triglyceride (*p* < 0.001) levels compared to both baseline and glipizide [17]. Second, the gut microbiota in the glipizide group had no significant changes. Conversely, treatment with acarbose resulted in significant changes of 69 mOTUs compared to baseline (Table 2). The overall diversity had significantly decreased at the end of the trial as indicated by lower gene count and Shannon index. Su et al. achieved similar results as well as a significant increase in *B. longum*, which directly correlated with HDL-C improvement [16]. Significant baseline decrease of *E. faecalis* was also reported.

Metformin resulted in significantly lower baseline anthropometric, glycemic, and lipid outcomes in both RCTs [18,19]. Compared to control groups, significant serum glucose and HbA1c improvements were found only by Wu et al. (Table 2) [19]. Changes in composition of gut microbiota were more evident than treating with acarbose (Table 2). Tong et al. reported a significant increase in five different genera from *Firmicutes*, *Proteobacteria*, and *Fusobacteria* phyla (Table 2) [18]. After the metformin treatment, the intestinal microbiome was significantly more diverse as indicated by the increased Simpson index, principal coordinate analysis (PCoA), and principal component analysis (PCA). Among the co-abundant group (CAG) clusters that were enriched in the metformin group, some had significant inverse correlations with glycemic control measurements. CAG 21 containing ten OTUs and CAG 25 containing four OTUs were related to lower HbA1c and fasting blood glucose (FBG) values, respectively. Furthermore, two CAGs that were lowered in metformin group had a significant correlation with alleviation of hyperglycemia and negative correlation with HOMA-β [18]. Wu et al. reported a significant negative correlation between *B. adolescentis* and HbA1c, which increased after metformin treatment [19]. *A. muciniphila* had also significantly increased in the metformin group, although there were no other correlations between any of the 81 altered strains.

### 3.2. Surgery as Anti-Diabetic Treatment

Three RCTs investigated the capabilities of bariatric surgery to alter gut microbiota and treat T2D [20,21,22]. The efficacy of Roux-en-Y gastric bypass (RYGB) was evaluated in all trials, while Murphy et al. and Lee et al. additionally analyzed sleeve gastrectomy (SG) and adjustable gastric banding (AGB), respectively [21,22]. All trials reported significant changes in anthropometric measurements 9–12 months after intervention (Table 2). Compared to baseline, significant metabolic outcome was present in only two trials with a decrease in HbA1c in both [21,22]. Murphy et al. additionally reported significantly lower HOMA-IR after RYGB and SG (*p* < 0.05).

Shifts of gut microbiota composition in analyzed RCTs are less straightforward than glycemic and anthropometric results (Table 2). In two trials, an increase in *Actinobacteria* abundance and lower HbA1c was present after RYGB [21,22]. Lee et al. also found that *Proteobacteria* had increased after both RYGB and AGB [22]. Other phyla changes are sporadic or inconsistent. In the trial by Murphy et al. the *Bacteroidetes* prevalence decreased after RYGB and increased after SG [21]. Although Cortez et al. investigated RYGB, the results were opposite [20].

At genera level, the increase in *Akkermansia* was consistent throughout studies [20,22]. However, only *Faecalibacterium* has increased significantly after RYGB compared to the control group. Furthermore, Lee et al. reported additional changes in 11 and 4 OTUs after RYGB and AGB, respectively [22]. *R. intestinalis* alterations did not differ between two types of surgery and had no significant correlation with glycemic control [21]. In fact, significant BMI, HDL, and other alterations were related to genera, which were unaffected by applied interventions. Cortez et al. reported a significantly higher Shannon index, although no individual species were identified nor glycemic response was achieved (Table 2) [20]. In other RCTs, a significantly more diverse microbiome compared to baseline characteristics was present after RYGB [21,22]. While these results did not differ from the control group, microbiota diversity after AGB was decreased significantly more compared to both RYGB and control groups. Neither intervention had any notable beta diversity results [22].

### 3.3. Probiotics

Metabolic and microbiome-changing effects of probiotics were analyzed in four RCTs [23,24,25,26]. All changes occurred in either the genera or species level or both. There were no significant anthropometric differences after interventions except in one trial, which found both a baseline improvement of anthropometric measurements as well as a significant increase in *L. reuteri* [23]. The most prevalent metabolic changes were observed in glycemic control, HbA1c specifically. The decrease was significant in two RCTs (*p* < 0.05; *p* = 0.0212), both of which had an increase in *Bifidobacterium* (*p* < 0.05; *p* = 0.049) [24,26]. One RCT found only a baseline change of HbA1c [25]. Considering insulin secretion, two RCTs indicated a baseline increase in insulin sensitivity index (ISI) and a significant decrease in fasting insulin levels (*p* < 0.05) [23,24].

Lipid profile improved in two RCTs [25,26]. In one of them, most of the differences were significant between groups [26] as the other provided only baseline results [25]. Although only one of the *L. reuteri* subtypes (ADR-1 and ADR-3) increased significantly (*p* = 0.017; *p* = 0.055), it still resulted to different outcomes. ADR-1 corresponded with significantly decreased cholesterol (*p* < 0.0467) and baseline LDL levels while the ADR-3 strain with reduced IL-1β concentrations (*p* = 0.0181). Moreover, ADR-3 group has significantly improved clinical parameters such as systolic blood pressure (*p* = 0.0248). To summarize, essential metabolic changes were more prominent in RCTs, in which *Bifidobacterium* abundance had increased.

### 3.4. Prebiotics

Due to prebiotic intervention heterogeneity, five RCTs provided inconsistent results [18,27,28,29,30]. The only trials that achieved clinically apparent improvements used fiber-containing prebiotics [27,28]. After 12 weeks, although indicated by different parameters such as HbA1c (*p* < 0.0001), area under the curve (AUC) (*p* < 0.05), or glucose effectiveness at zero insulin (GEZI) (*p* = 0.0212), glycemic control within the intervention groups was significantly better [27,28]. Moreover, although specific microbiome alterations in these trials did not coincide, both resulted in higher Shannon, inverse Simpson indices, or species richness values. The difference between groups was significant only in one of them, which interestingly corresponded to changes in lipid metabolism indicated by significantly lower triglyceride (*p* < 0.01), total cholesterol, and LDL concentrations (*p* < 0.001) [27]. Only one trial, which added an AMC Chinese herb formula to a preexisting metformin treatment regimen, achieved baseline anthropometric, glycemic, and lipid metabolism improvements [18]. In the AMC formula group, there also was a significantly lower HOMA-IR (*p* < 0.05) and triglyceride concentrations (*p* < 0.01). Most importantly, beta diversity in the AMC formula group was higher as well. Other RCTs found only sporadic alterations, indicated in Table 2 [29,30].

### 3.5. Synbiotics

Zhang et al. compared the effects between berberine, probiotics, and combined therapy, known as synbiotics [31]. Compared to placebo, the latest provided the most significant metabolic changes, including improved HbA1c (*p* < 0.001), other glycemic control markers, both HOMA-IR (*p* = 0.01) and HOMA-β (*p* = 0.02), and lipid levels in all age groups. In the berberine arm, HOMA-β improvement (*p* < 0.001) was only baseline while HOMA-IR did not change. In the probiotics arm, there was no positive effects related to glycemic control or insulin secretion, except for lower triglyceride (*p* = 0.04) and higher HDL (*p* = 0.05) concentrations. The microflora alterations corresponded to the extent of the metabolic change. Bacteria gene count (*p* < 0.001), Shannon index (*p* < 0.05), and principal coordinate analysis (PCoA) (*p* < 0.001) have significantly decreased in the berberine and combined intervention groups only. Moreover, the most noticeable changes within the species level have occurred in these very groups, 70 and 80 species, respectively.

### 3.6. Changes in Composition of Intestinal Microbiome

#### 3.6.1. Phylum Level

Considering phyla changes, metabolic alterations most frequently occurred in three groups, *Firmicutes*, *Bacteroidetes*, and *Actinobacteria* (see Appendix A). According to thirteen RCTs, quantitative increase in *Firmicutes* was simultaneously present with a positive anthropometric and metabolic response [17,18,19,20,21,22,23,24,25,26,27,28,29,31], and two RCTs provide contradicting results (see Appendix A) [16,30]. Tong et al. found positive metabolic and anthropometric outcomes in patients with both increase and decrease of genera from *Firmicutes* phylum after the two types of treatment. Changes of *Actinobacteria* and *Bacteroidetes* were almost equally frequently common among analyzed RCTs eleven and nine, respectively. The change of abundance of *Bacteroidetes* was opposite to *Firmicutes*. Seven RCTs indicate that a decrease in the prevalence of this phylum was present with positive anthropometric or metabolic results [17,18,20,21,27,29,31]. Prominently, all the trials that resulted in decrease in *Bacteroidetes* abundance also had an expansion of *Firmicutes* (see Appendix A). However, seven RCTs found an improvement of metabolic parameters with a higher abundance of *Bacteroidetes* at the end of the trial [17,18,19,20,21,30,31]. Several trials reported overall contradictory findings with both lower *Firmicutes* and higher *Bacteroidetes* prevalence [17,18,19,30,31].

Increase in *Actinobacteria* overlapped with expansion of *Firmicutes* in eight out of ten RCTs (see Appendix A). More importantly, higher *Actinobacteria* abundance was parallel to better glycemic control or improved lipid profile at follow-up [16,17,19,21,22,24,26,27,28,31]. In contrast, two trials resulted in simultaneous increase and decrease in genera or species from *Actinobacteria* alongside improved glycemic control [17,31].

Changes in less frequently found phyla such as *Proteobacteria*, *Verrucomicrobia*, *Fusobacteria*, *Euryarchaeota*, and *Spirochaetes* have also occurred (see Appendix A) [17,18,19,20,22,23,27,29,30,31]. Even though *Proteobacteria* increased in five [18,19,22,30,31] and decreased in four RCTs [17,18,19,31], all of them reported better glycemic control at follow-up. Gu et al. reported adjacent decrease in anthropometric measurements and lipid profile with only a decline in *Proteobacteria* [17]. Consistency and genera variability were two main distinctions within other rare groups. Results regarding higher *Verrucomicrobia* prevalence were unanimous [20,22,27,29]. In contrast to the three main phyla, the increase in *Verrucomicrobia* and *Fusobacteria* was concordant with specific genera, *Akkermansia* and *Fusobacterium,* respectively [18,19,20,22,27,29,31].

#### 3.6.2. Genus Level

Four most occurring alterations within the *Firmicutes* were *Lactobacillus*, *Faecalibacterium*, *Clostridium*, and *Roseburia* [17,18,19,21,22,23,24,25,26,27,29,31]. Despite a few contradictory results, *Lactobacillus* [17,19,23,24,25,26,29,31] and *Faecalibacterium* [18,22,27,31] increased, while prevalence of *Clostridium* genus [17,18,19,29,31] declined. Regarding *Roseburia*, both increased [21,22] and lowered [17,31] abundance post-treatment were present. Some microbiota changes were provided as clusters (e.g., *Clostridium*, *Lactobacillus*, and *Enterococcus*) [17,18,25,29]. Changes among other less frequent genera were inconsistent (see Appendix A).

From nine RCTs, no specific tendencies within *Bacteroidetes* were observed as *Bacteroides*, *Prevotella*, and other less common genera either increased or decreased [17,18,19,20,27,29,30,31]. The only exception was *Alistipes,* which declined in three RCTs without conflicting reports [17,18,31]. *Bifidobacterium* attributed to most of *Actinobacteria* alterations. The microbiome alterations regarding these genera were predominantly positive [16,17,19,24,26,27,28,31].

Among the less frequent phyla, *Verrucomicrobia* and *Akkermansia* abundance increase was an exclusive alteration [20,22,27,29]. *Fusobacterium* represented results in *Fusobacteria*, although the changes were only roughly concordant [17,18,19,31]. The alterations in the *Proteobacteria* phylum had less dominance by specific genera, except for *Escherichia*, which increased in several RCTs [17,18,19,22,30,31]. The specific changes regarding species were scarce (see Appendix A).

#### 3.6.3. Changes in Diversity

More complex microbiome evaluation was analyzed in over half of the RCTs [17,18,19,20,21,22,23,27,28,29,31]. All trials that resulted in biodiversity shifts had a positive anthropometric and/or metabolic response (Table 3). Different diversity indexes were mostly used complimentary to one another. Medina-Vera et al. was the exception as only Shannon index was used [27]. Two more RCTs reported results of increased Shannon index suggesting that the applied treatment resulted in a more complex microbiome [20,28]. In these trials Species richness or Chao1 estimator had also increased. Results by other two RTCs reported higher Simpson and Chao1 indexes as well [18,22]. Although Pedersen et al. reported that both Shannon index and species richness had increased, inverse Simpson index was also higher [28]. In addition, three more trials correspond to such contradictions, as decreased microbiome diversity was adjacent to better glycemic control [17,18,31].

Beta diversity was evaluated in seven RCTs; however, the majority had no significant results post intervention (Table 3) [18,19,20,22,27,29,31]. Principal coordinate analysis (PCoA) was implemented in all the listed trials, while principal component analysis (PCA) was used only once [18]. In this study, PCA and PCoA results based on Bray–Curtis distance confirmed significant microbiome differences after the intervention. The difference remains after evaluating phylogenetics with both weighted and unweighted UniFrac distances. There was no significant post-interventional clustering in other studies that implemented UniFrac [20,22]. PCoA based on Bray–Curtis distance was implemented most frequently, and only one more trial reported significant results at follow-up [31].

### 3.7. The Overall Significant Changes after All Types of Treatment

Across six most frequently altered phyla, higher abundances of *Firmicutes* and *Proteobacteria* were the most consistent results as four intervention types achieved such changes. *Bacteroidetes* and *Verrucomicrobia* increased after surgery and prebiotic implementation, while *Actinobacteria* increased after surgery and probiotics. Bariatric surgery, which is of highest efficacy in weight loss and T2D remission induction, had the highest number of changed phyla, especially after RYGB. Prebiotic supplementation was second, and probiotics and oral anti-diabetic medication were third and resulted in higher abundance of two separate phyla. However, all these tendencies are relative because they are based on majority count and taxonomic hierarchy (Table 4).

## 4. Discussion

This is the first systematic review that included randomized controlled human trials investigating conventional glucose-lowering treatment or bariatric surgery, or probiotic, prebiotic, or symbiotic supplementation effects on both glycemic control and gut microbiome in T2D patients exclusively. A total of 16 eligible studies involving 1301 participants were reviewed. The most common alterations were increased abundance of *Firmicutes* and *Proteobacteria* parallel to improved glycemic control. Bariatric surgery, especially Roux-en-Y gastric bypass, had the highest variety of changed bacteria phyla. Lower diversity post-treatment was the most significant biodiversity result.

### 4.1. Changes in Composition of Intestinal Microbiome

In this systematic review, the most frequent changes at the phylum level were present among three groups, *Firmicutes*, *Bacteroidetes*, and *Actinobacteria*. According to metagenomic studies, *Firmicutes* and *Bacteroidetes* represent 90% of intestinal microflora in healthy humans [32,33]. In T2D subjects, the same phyla represented the majority, although *Firmicutes* abundance was significantly lower in T2D individuals [3,33]. Larsen et al. determined the *Bacteroidetes*-to-*Firmicutes* ratio, which positively correlated with plasma glucose, changed from 1:1.6 to 1.4:1 in T2D subjects [33]. In the present review, *Firmicutes* and *Bacteroidetes* prevalence changes after anti-diabetic treatment could represent flora of non-T2D individuals, which might have also influenced improved glycemic control.

Across analyzed RCTs, higher *Lactobacillus* genus abundance post-treatment was the most common change in the *Firmicutes* phylum [17,19,23,24,25,26,29,31]. Regarding *Lactobacillus* genus prevalence differences between T2D and non-T2D individuals, previous studies reported heterogenous results [4,34,35,36]. It is hypothesized that *Lactobacillus* positive effects on glucose homeostasis and insulin sensitivity stems from bacteria-mediated butyrate production that acts as a metabolic modulator [2,36,37]. Moreover, butyrate producing *Faecalibacterium prausnitzii* and *Roseburia intestinalis* that tend to be less abundant in T2D subjects increased in several RCTs [21,22,25,26,31]. This proposes that anti-diabetic treatment could stimulate the growth of depleted bacteria, which enhances the metabolic response.

Regarding another genus from the *Firmicutes* phylum, the decline in *Clostridium* abundance after T2D treatment was more prevalent, although glycemic control improved in most included trials [17,18,19,25,29,31]. *Clostridium* genus has substantial variability at species level as it can represent up to 95% of the *Firmicutes* phylum [33,38]. Prominently, the respective genus does not define all properties, and individual species can influence metabolic pathways by producing different substances [39,40]. In this review, the decline in the *Clostridium* genus that appeared opposite to overall changes of the *Firmicutes* phylum could have corresponding positive influence on glucose metabolism with other genera.

*Actinobacteria* was the second most frequently shifted phylum in included trials, while *Bifidobacterium* represented most changes [16,17,19,24,26,27,28,31]. Contrary to the *Firmicutes* phylum, *Actinobacteria* prevalence between healthy and T2D subjects is similar [3,38,41]. *Bifidobacterium* is frequently regarded as the source of protective bacteria against T2D [4]. It can influence glucose homeostasis by promoting gene expression of insulin signaling pathways and producing short-chain fatty acid (SCFA) precursors [42,43]. SCFA-induced toxicity can result in selective inhibition of pathogenic bacteria, which allows beneficial microbiomes to flourish [44]. In analyzed RCTs, increased *Actinobacteria* abundance was frequent with higher *Firmicutes* prevalence [17,19,20,21,24,26,27,31]. It is not clear whether *Actinobacteria* or *Firmicutes* have more significant effects on glycemic control. Either phyla could act as a catalyst to anti-diabetic treatment that induces the shift towards non-T2D microbiome and better glycemic control.

Regarding the last most altered phylum [17,18,19,20,21,27,29,30,31], researchers suggest that *Bacteroidetes* depletion would benefit glycemic control, as higher abundance of this phylum is linked to worse glucose homeostasis [3,4,38]. Although increased prevalence of *Bacteroidetes* in RCTs was more common, glycemic control improvements were more frequent when abundance decreased [19,20,30,31]. Zeevi et al. also found that the abundance of *Bacteroides* increased parallel to positive outcomes after applying a machine learning generated personalized diet [44]. Other genera shifts could be the cause of *Bacteroides* heterogeneous results post-treatment. According to Johnson et al., there is a binary enterotype concept between *Bacteroides* and *Prevotella* genus, meaning that the subject’s microbiome predominantly comprises either one genus or the other [45]. In a meta-analysis, low *Bacteroides* abundance coexisted with high *Prevotella* prevalence [46]. In this review, such tendencies were observed only once [17,19,30,31].

Previous studies found that within the *Bacteroidetes* phylum, *Alistepes* genus abundance is significantly increased in cases of T2D [2]. In analyzed RCTs, lower post-treatment prevalence could represent the shift towards a non-T2D microbiome [17,18,31]. Metformin and antibiotics could have influenced these results [4,17,18,33]. RCT by Zhang et al. was an exception, which included drug-naive subjects [31].

Several authors identified *Escherichia* to be significantly more prevalent in T2D subjects, suggesting that glycemic control could improve upon bacteria abundance reduction [2,3]. In this review, among *Proteobacteria* phyla, the most frequent shifts were an increased prevalence of *Escherichia* genus and *E. coli* species [18,19,30], which was also present among drug-naive subjects [31]. Metformin usage in RCTs or bias related to study design could account for opposite results reported in previous studies [4,18,19,30,47,48]. The positive effect on glycemic control despite increased prevalence of *Escherichia* genus could be associated with lactose-consuming *Lactobacillus* bacteria, which is enriched parallel to *E. coli* growth [49]. In a carbohydrate-specific environment (e.g., low in lactose), *E. coli* possesses an ability to maximize growth by suppressing unabundant carbohydrate catabolism and thriving on remaining sugars [50].

Among less frequently changed phyla, increased *Akkermansia* prevalence and parallel metabolic improvement post-treatment [22,27,29] was observed. However, there is some contradiction regarding the prevalence of *Akkermansia,* as some studies report lower abundance in patients with pre-diabetes and T2D [4,49], while others find higher *Akkermansia* prevalence in T2D patients [2,51]. More evidence links *Akkermansia* to obesity rather than diabetes, as the intestinal flora of overweight individuals had significantly less *A. municiphila,* which increased upon weight reduction [52]. Moreover, glycemic control improvements were more significant if an obese subject had higher baseline *Akkermansia* abundance [53]. Among RCTs, *Akkermansia* increase was observed after bariatric surgery and dietary interventions [20,22,27,29]. However, some study participants were treated with metformin, which is known to promote *A. municiphila* growth [4,52,54]. There is lack of evidence regarding other oral antidiabetic drug abilities to alter *A. municiphila* abundance [55]. Therefore, to interpret the effect of weight loss on *Akkermansia* abundance, treatment with metformin must be taken into consideration.

Low-grade intestinal inflammation with increased permeability characterizes both obesity and T2D. *A. municiphila* was proven to have a role in the epithelial barrier function, suggesting that it could improve metabolic outcomes in T2D patients. Aron et al. concluded that *A. municiphila* influences the improvement of inflammation, insulin resistance, and glycemia in cases of obesity and diabetes [56], yet the relationship is not linear [57]. Insulin resistance lowering effects could be advantageous not only in T2D control, but in lowering risk of cardiovascular diseases as well [58]. In fact, Schneeberger et al. found that *Akkermansia muciniphila* was inversely associated with cardiovascular risk [57].

### 4.2. Changes in Diversity

In trials that reported increased diversity, other diversity measurements contradicted one another, or no glycemic control improvement was observed [20,27,28]. Pedersen et al. reported conflicting increase in both Shannon index (equally weights the number of different taxa observed in the community and the equitability of the taxa frequencies) [59] and inverse Simpson index. The latter provides more weight to equitability of the taxa frequencies in a community making it less sensitive to rare species, suggesting that less abundant *Bifidobacteria* post-treatment had not affected overall diversity [28,59,60]. Growth of *Roseburia*, *Faecalibacterium*, *Akkermansia* genera, and *Proteobacteria* phylum, resulted in increased observed richness (number of different taxa in a sample) and phylogenetic diversity, implying that anti-diabetic treatment resulted in higher number of taxonomically distant bacteria [22,59]. However, the bacteria that have increased were already present, which suppressed growth of rare species as indicated by the decreased Chao1 index, which evaluates the abundance of each taxon. Therefore, anti-diabetic treatment is unlikely to produce a significantly more diverse microbiome.

The decrease in Shannon index supported by lower bacteria gene count and other indexes were found together with improved glycemic control post-treatment in analyzed RCTs [17,31]. According to Le Chatelier et al., insulin resistance based on HOMA-IR is considerably lower in healthy individuals that had a low bacteria gene count [59]. Compared to healthy subjects, Shannon index did not differ in observational studies, implying the significant effect on glucose homeostasis stems from a specific species [61,62]. In our review, significantly different microbiome composition, as indicated by significant results in beta diversity that compared microbiome before the applied treatment and after, was present only with lower diversity. Thus, the most prominent metabolic improvements are likely to be reached when anti-diabetic treatment promotes corresponding bacteria shifts towards a healthy individual’s microbiome.

### 4.3. Oral Anti-Diabetic Treatment

Human studies analyzing anti-diabetic medication effects on gut microflora make up only a small part of all research [63]. Forslund et al. found that treating T2D patients with metformin significantly increases *Escherichia* and reduces abundance of *Intestinibacter*. In the present review, Wu et al.’s study had similar results, and it was the only trial in the medication group that achieved significantly improved glycemic control [17,64]. On the other hand, metformin-untreated T2D was associated with a decrease in *Subdoligranulum*, *Roseburia*, and a cluster of butyrate-producing *Clostridiales*. A decrease in *Roseburia* was observed in groups that implemented α-glucosidase inhibitors, prebiotics, or synbiotics [17,31] after bariatric surgery [17,21,22,31]. It is unclear whether the effects were achieved by beneficial microflora or via treatment-specific mechanisms.

Studies highly suggest that metformin–microflora interactions are a two-way process. Elbere et al. and Sun et al. found that metformin clinical benefits are partly mediated by bacteria-specific mechanisms such as glucose-SGLT1-sensing glucoregulatory pathway associated with *Lactobacillus* increase or *B. fragilis*—glycoursodeoxycholic acid (GUDCA)—intestinal farnesoid X receptor (FXR) [65,66]. Furthermore, higher abundances of *Enterococcus faecium*, *Lactococcus lactis*, *Odoribacter*, and *Dialister* were linked to a better response to metformin, while higher *Prevotella copri* suppresses it [65]. Based on these studies, microbiome preparation with proven gut-modulating interventions could improve response to first-line treatment.

Benefits of acarbose in T2D are suggested to be associated with increased abundance of SCFA-producing taxa such as *Faecalibacterium*, *Prevotella*, and *Lactobacillus*, which were correlated with decreased postprandial insulin secretion and lower triglyceride levels [67]. In this systematic review, acarbose did improve glycemic control and lipid metabolism in both RCTs, although the prevalence of the beforementioned genera (*Prevotella* and *Lactobacillus*) increased only in one of them [17]. It is important to note that Su et al. concentrated on acarbose effects on *Bifidobacterium longum* specifically, suggesting that changes might have occurred but not been reported [16]. Drug dose and treatment continuity are also crucial as Forslund et al. provided evidence that only higher doses of acarbose resulted in substantial microbiome alteration [64]. After cessation, the microbiome quickly shifted back to mirror the control group, suggesting that consolidation can only be achieved by continuous treatment. More research is necessary to understand effects of metformin and acarbose on human microbiome, how these drugs modulate the gut, and if any synergistic interactions are possible.

Evidence from antidiabetic medication trials such as GLP-1 agonists clearly states that an antidiabetic drug not only provides glycemic control, but also reduces cardiovascular disease risk via complex mechanisms. GLP-1 analogues decrease blood pressure and reduce total cholesterol, LDL, and TG levels, which partially can be attributed to weight loss [58]. Although the underlying mechanisms are not completely understood, Zhao et al. propose that incretin augmentation with GLP-1 agonists delays gastric emptying and gut transit time, which in turn alter factors that are known to affect the composition of the microbiome such as local pH values and nutrient composition [68]. Authors also state that microbiome modulation including inhibition of microbiota abundance and diversity and elevation of the *Bacteroidetes/Firmicutes* ratio can prevent weight gain, which partially overlaps with the results of our analysis. However, Madsen et al. found opposite results as liraglutide-treated mice gut dominated by *Firmicutes* related species such as *Clostridiales spp*., *Lachnospiraceae*, and *Enterococcus faecium* correlated to weight loss and improvements in glucose tolerance and lipid levels [69]. Although researchers did not achieve significant bacterial abundance reduction, *Firmicutes* increase is also one of the main findings regarding human T2D trials. Unfortunately, most microbiome research regarding antidiabetic medication is still in the animal trial phase [64]. More studies are warranted to establish whether their therapeutic uses are partially mediated through the gut microbiome alterations and whether such changes are relevant to humans.

### 4.4. Surgery as Anti-Diabetic Treatment

According to various authors, duodeno-jejunal exclusion effectively modulates both insulin resistance and secretion [70,71]. In this review, out of three trials, RYGB only once resulted in significantly lower HOMA-IR [21]. Among included trials, HbA1c had more compliance, which could be explained by glucose absorption alterations and faster gastric emptying [71]. Body weight decreased in all studies, which was the main cause of T2D remission [70]. However, several weight loss-independent mechanisms, such as microbiome changes, might have influenced glycemic control [72].

Multiple systematic reviews and meta-analyses identified *Proteobacteria* and *Bacteroidetes* to frequently increase, while *Firmicutes* phylum tended to decrease after bariatric surgery [9,11,71,73]. Gou et al. stated that the increase in *Bacteroidetes* and decrease in *Firmicutes* had a strong level of evidence [9]. *Faecalibacterium* abundance post-surgery was lower in various studies as well [9,11,71,74]. In this review, trials reported contradicting results [20,21,22]. Trial subjects’ heterogeneity is the most likely explanation, as inclusion criteria of other reviews were not limited to T2D subjects and had individuals with different comorbidities [9,11,71]. In this study, SG promoted the growth of *Bacteroidetes*, while RYGB decreased it as in other RCTs [20,21,22]. Heterogenous microbiome changes may stem from structural or functional gut differences after bariatric surgery [73]. Moreover, lower fat and higher complex carbohydrate intake or calory restrictions post-RYGB could also explain the alterations in the microbiome [75,76].

### 4.5. Probiotics, Prebiotics, and Synbiotics

Although yet unproven, *Lactobacillus* and *Bifidobacterium* show promise in a variety of disorders through suppressing growth or invasion of pathogenic bacteria, improving intestinal barrier function, and modulating the immune system [77,78,79,80]. According to Gurung et al., both are among bacteria negatively associated with T2D together with *Akkermansia*, *Bacteroides*, *Faecalibacterium*, and *Roseburia* [2]. A recent meta-analysis concluded that probiotics are beneficial to use to improve glycemic control in T2D [81]. All studies included in the meta-analysis implemented at least one *Lactobacillus* strain while some used combinations with *Bifidobacterium* or other genera. RCTs in this systematic review followed the same principles [25,26,31]. Notably, authors of the PREMOTE trial indicate that most studies have essential methodological limitations, heterogeneous target populations, and use different probiotic strains, making it difficult to make conclusions about supplementation efficacy [31,82,83]. Within the same trial, isolated probiotic intervention did not achieve any positive metabolic alteration compared to isolated prebiotic and synbiotic groups, suggesting a possible synergistic effect.

Although some authors indicate that 6 weeks is a cut-off point when probiotics take effect, most significant alterations were observed after at least 8 weeks [81,84]. More than half of analyzed RCTs implemented probiotics for at least 12 weeks and resulted in improved glycemic control, suggesting that previous trials might be ended prematurely [24,25,26]. Regarding microbiome changes, all trials resulted in the promotion of bacteria related to the probiotic formula. In the study by Shin et al., an increased *Lactobacillus* genus was observed together with *Clostridium* species growth, which returned to primary values when the intervention was discontinued, strengthening the importance of gut-modulation continuity [25]. Similar *Clostridium* species level changes were observed in other RCTs when *Lactobacillus*, especially *L. casei* and *L. gasseri,* had increased [19,31]. Such alterations did not occur with any *Bifidobacterium*-containing formulas, suggesting that probiotic composition efficacy is closely related to a specific disease. This disease-specific effect hypothesis is supported by the trial of Hsieh et al., in which only live *L. reuteri* ADR-1 had a significant positive effect on glycemic control and lipid profile, while the heat-killed *L. reuteri* ADR-3 resulted in inflammatory marker alterations [26]. Thus, a research-based agreement on specific probiotic formulas and study protocols used in T2D related trials is warranted.

Fructooligosaccharides (FOS), galactooligosaccharides (GOS), and inulin are known to have bifidogenic properties and are among the most frequently analyzed prebiotic supplements [85]. Inulin-type carbohydrates increase the density of PYY-producing cells, thus showing its role in reducing appetite and food intake and enhancing obesity treatment [86]. A meta-analysis by Rao et al. confirms prebiotic effectiveness in reducing insulin resistance but states that the main mechanism is likely related to microflora abundance alterations [87,88]. Isolated GOS interventions and FOS combinations with inulin improved glycemic control and lipid profile levels, which matches the results of trials within our analysis [18,27,28,29,31,89,90,91,92]. However, it is crucial to note the differences in study populations, as FOS was used in T2D studies while GOS was implemented in obesity trials [93]. Using xylooligosaccharide lowered OGTT 2 h insulin levels in adults with prediabetes, suggesting that these prebiotics are more beneficial in alleviating the risk factors of T2D than treating it [94]. It is unclear whether efficacy of certain prebiotics is universal or related to a specific disease.

The lack of standardization is prevalent in prebiotic trials. In this review, only one study implemented isolated GOS while other trials used various prebiotic combinations. All resulted in increased abundance in genera that are negatively associated with disease, suggesting that the effects could be attributed to specific compounds within them [2,81,95]. RCTs that resulted in most significant glucose and lipid metabolism improvements also had significantly lower diversity [18,31]. Even though other trials found that Shannon index had increased, a statement that these trials achieved higher microbiome diversity would be premature [27,28].

## 5. Limitations

The foremost limitation of estimations is that subjects, study designs, applied intervention, and the outcome definitions differed among included trials, lowering the generalizability of this review. Applicability is also influenced by incomplete data in some trials or of certain results. Furthermore, the validity of the studies varied. Only two trials had low overall risk-of-bias, while high risk was a dominant feature. Highest bias risk was found in deviations from intended interventions.

## 6. Conclusions

This systematic review analyzed sixteen randomized controlled trials and reported commonly found microbiome changes alongside metabolic outcome as well as provided insights behind possible causes. Anti-diabetic treatment induced the growth of depleted bacteria within phyla, genera, and species levels that were observed together with improved glycemic control. In some cases, such changes shifted the gut microbiome towards the flora of healthy individuals. A substantial number of microbiome changes could also be explained by factors such as intervention technique, prior treatment status, and primary microbiome composition. On a such delicate topic, where there are more questions than answers, the most beneficial direction of research would be developing strategies that most effectively increase the levels of depleted gut bacteria, lower diversity, and, eventually, reach a non-T2D microbiome. Meta-analyses that would quantifiably evaluate microbiome changes after T2D treatment are also needed.

## Figures and Tables

**Figure 1 medicina-57-01084-f001:**
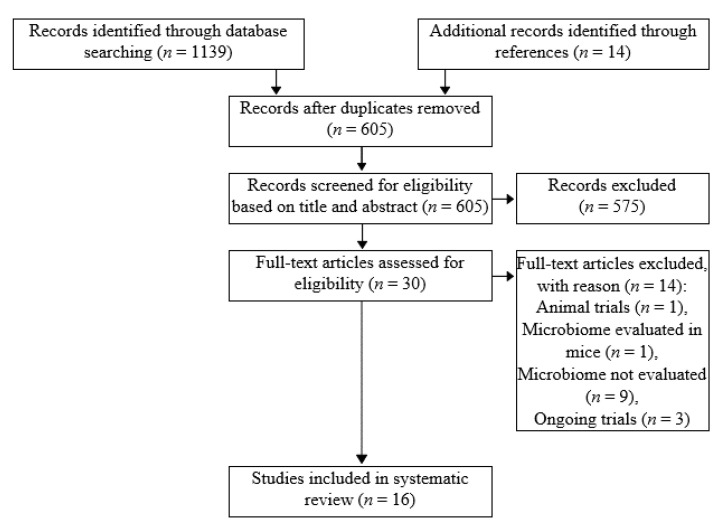
PRISMA flow diagram detailing the inclusion process.

**Table 1 medicina-57-01084-t001:** General view of included trials and risk-of-bias evaluation.

Source	No. of Subjects (Male)	Intervention Group	Control Group	Follow-Up	R	D	Mi	Me	S	O
Su et al. [16]	95 (46)	T2D treatment with acarbose	T2D treatment without acarbose	4 weeks						
Gu et al. [17]	94 (58)	Acarbose	Glipizide	12 weeks						
Tong et al. [18]	200 (100)	Metformin with prebiotics (Chinese herbal formula)	Metformin without prebiotics	12 weeks						
Wu et al. [19]	40 (17)	Metformin	Placebo	8 and 16 weeks						
Cortez et al. [20]	21 (unspec.)	Surgery (DJB)	Optimal T2D treatment (metformin 2 g/day, gliclazide 30 mg)	24 and 48 weeks						
Murphy et al. [21]	14 (8)	Surgery (SG)	Surgery (RYGB)	48 weeks						
Lee et al. [22]	12 (0)	3-arm study: surgery (AGB or RYGB)	Medical weight loss	Variable						
Mobini et al. [23]	53 (35)	Probiotics (*L. reuteri*)	Placebo	12 weeks						
Firouzi et al. [24]	129 (67)	Probiotics (multi-strain)	Placebo	6 and 12 weeks						
Sato et al. [25]	68 (49)	Probiotics (*L. casei*)	Placebo	8 and 16 weeks						
Hsieh et al. [26]	68 (38)	3-arm study: probiotics (*L. reuteri* ADR-1 or ADR-3)	Placebo	12, 24, and 36 weeks						
Medina-Vera et al. [27]	53 (19)	Prebiotics (non-specific functional foods)	Placebo	12 weeks						
Pedersen et al. [28]	29 (29)	Prebiotics (GOS)	Placebo	12 weeks						
Shin et al. [29]	12 (9)	Prebiotics (*Scutellaria baicalensis*)	Placebo	8 weeks						
Balfego et al. [30]	32 (15)	Standard T2D diet with prebiotics (non-specific functional foods)	Standard T2D diet without prebiotics	24 weeks						
Zhang et al. [31]	381 (245)	4-arm study: probiotics (multi-strain) and/or prebiotics (berberine)	Placebo	13 weeks						

R—bias arising from randomization process; D—bias due to deviations from intended interventions; Mi—bias due to missing outcome data; Me—bias in measurement of the outcome; S—bias in selection of reported results; O—overall bias. Low bias—green; some concerns—yellow; high risk—red. T2D—type 2 diabetes; AGB—adjustable gastric banding; DJB—duodenal–jejunal bypass; GOS—galacto-oligosaccharides; RYGB—Roux-en-Y gastric bypass; SG—sleeve gastrectomy; T2D—type 2 diabetes, unspec.—unspecified.

**Table 2 medicina-57-01084-t002:** Intervention-specific alterations in microbiome composition, biodiversity, and anthropometric or metabolic improvements.

Applied Intervention	RCT	Changes in Microbiome Composition			Changes in Microbiome Diversity		Anthropometric or Metabolic Improvements					
		Phylum	Genera	Species	Alpha Diversity	Beta Diversity	A	G	I	C	L	
Oral antidiabetic pharmaceuticals	Su et al. [16]	–	–	↑: *B. Longum*† ↓: *E. faecalis*	–	–	–	HbA1c, [glucose]	–	–	Ch, TG, LDL	
	Gu et al. [17]	–	–	(Acarbose) ↑: 36 mOTUs ↓: 33 mOTUs	↓: Rarefaction, gene count, Shannon	–	BW, BMI	HbA1c, [glucose]	AUC	HOMA-IR	Ch, TG	
		–	–	(Glipizide) No significant results	No significant results	–	–	HbA1c, [glucose]	–	–	–	
	Tong et al. [18]	–	↑: *Megamonas*, *Escherichia/Shigella*, *Klebsiella*, *Blautia*, *Fusobacterium* ↓: *Alistipes*, *Bacteroidetes*	–	↑: Simpson	↑: Bray-Curtis distance PCA, PCoA	BW, BMI	HbA1c, [glucose]	–	HOMA-B	Ch, LDL, HDL	
	Wu et al. [19]	–	↑: *Escherichia*, *Bifidobacterium*, *Intestinibacter*	↑: 67 strains ↓: 18 strains	No significant results	No significant results	BMI	HbA1c, [glucose]	–	HOMA-IR, HOMA-B	HDL	
Surgery	Cortez et al. [20]	↑: *Bacteroidetes*, *Verrucomicrobia*	↑: *Bacteroides*, *Akkermansia*, *Dialister*	–	↑: Shannon†, Chao1 estimator, Simpson^§^	Unifrac distance: dispersion ‡	BW	–	–	–	–	
	Murphy et al. [21]	(RYGB) ↑: *Firmicutes*, *Actinobacteria* ↓: *Bacteroidetes*	–	↑: *R. Intestinalis*	↑: species richness	–	BMI	HbA1c	–	HOMA-IR	–	
		(SG) ↑: *Bacteroidetes*	–	↑: *R. Intestinalis*	–	–	BMI	HbA1c	–	HOMA-IR	–	
	Lee et al. [22]	(AGB) ↑: *Proteobacteria*	↑: *Akkermansia*	↑: 2 OTUs ↓: 2 OTUs	↓: observed species^¶^, Chao1^¶^, PD^¶^	Unifrac distance: no distant clustering	BMI	HbA1c	–	–	–	
		(RYGB) ↑: *Proteobacteria*, *Actinobacteria*	↑: *Faecalibacterium*†, *Akkermansia*	↑: 10 OTUs ↓: 1 OTUs	↑: observed species, Chao1, PD		BMI	HbA1c	–	–	–	
Probiotics	Mobini et al. [23]	–	–	↑: *L. reuteri*†	No significant results	No significant results	BW, BMI	–	ISI	–	–	
	Firouzi et al. [24]	–	↑: *Lactobacillus*, *Bifidobacterium*†	–	–	–	–	HbA1c	fasting	–	–	
	Sato et al. [25]	–	↑: *Lactobacillus*†, *Enterococcus*	↑: *C. coccoides*!†, *C. leptum*!†, *L. gasseri*, *L. casei*†, *L. reuteri*	–	–	–	HbA1c	–	–	Ch	
	Hsieh et al. [26]	–	↑: *Bifidobacterium*	↑: *L. reuteri*†	–	–	–	–	–	–	Ch	LDL
	Zhang et al. [31]	–	–	↑: 12 strains	↓: gene count	No significant results.	–	–	–	–	TG, HDL	
Prebiotics	Tong et al. [18]	–	↑: *Paraprevotella*, *Megamonas*, *Faecalibacterium*, *Klebsiella*, *Lachnospiraceae*, *Blautia* ↓: *Bacteroidetes*, *Parasutterella*, *Clostridiales*, *Alistipes*, *Clostridium*	–	↓: Rarefaction, Chao1	↑: Bray-Curtis distance PCA, PCoA. weighted and unweighted UniFrac distances†, abundance-weighted, binary Jaccard distances†	BW, BMI	HbA1c, [glucose]	–	HOMA-IR	Ch, TG, LDL	
										HOMA-B	HDL	
	Medina-Vera et al. [27]	–	–	↑: *F. prausnitzii*, *A. muciniphila*, *B. longum*, *B. fragilis* ↓: *P. copri*	↑: Shannon†	–	–	HbA1c, AUC	ISI	–	Ch, TG, LDL	
	Pedersen et al. [28]	–	↑: *Bifidobacterium*	–	↑: Shannon, inverse Simpson Richness	No significant results	–	GEZI	–	–	Ch, LDL	
	Shin et al. [29]	–	↑: *Lactobacillus*†, *Akkermansia*†, *Megamonas*, *Mobilitalea*, *Acetivibrio* ↓: *Bifidobacterium*†, *Clostridium*, *Oscilibacter*, *Alloprevotella*	–	No significant results	No significant results	–	AUC	–	–	–	
	Balfego et al. [30]	↓: *Firmicutes*	↑: *Bacteroides*, *Prevotella*	↑: *E. Coli*	–	–	–	–	fasting	HOMA-IR	–	
	Zhang et al. [31]	–	–	↑: 40 strains ↓: 30 strains	↓: gene count†^, Shannon	↑: Bray-Curtis distance PCoA†	–	HbA1c, [glucose], 2hPPG	–	HOMA-B	Ch, TG, LDL, HDL	
Syn.	Zhang et al. [31]	–	–	↑: 41 strains ↓: 39 strains	↓: gene count†^, Shannon†^	↑: Bray-Curtis distance PCoA†	–	HbA1c, [glucose], 2hPPG	–	HOMA-IR, HOMA-B	Ch, TG, LDL, HDL	

All changes listed are statistically significant compared to baseline (*p* < 0.05), unless specified otherwise (yellow cells). “–” means that a certain parameter was not evaluated in a specific trial or no significant changes were reported. †—statistically significant compared to control group (green cells; note: not necessarily all components of the cell had altered significantly). ‡—differs from control, which clustered. §—statistically significant change in the control group compared to intervention group. ¶—lower than in the control and alternate intervention groups (*p* < 0.05). ^—lower than in the probiotics group (*p* < 0.001). !—no statistical difference compared to baseline (*p* ≥ 0.05). Syn.—synbiotics; A—anthropometric; G—glycemic control; I—insulin secretion; C—combined values of glycemic control and insulin secretion; L—lipid profile. AGB—adjustable gastric banding; AUC - area under the curve; BMI—body mass index; BW—body weight; Ch—total cholesterol; GEZI—glucose effectiveness at zero insulin; HbA1c—glycated hemoglobin; HDL—high-density lipoprotein; HOMA-B (IR) —homeostatic model assessment of beta cell function (insulin resistance); ISI—insulin sensitivity index; LDL—low-density lipoprotein; mOTU—molecular operational taxonomic unit; OTU—operational taxonomic unit; PCA—principal component analysis; PCoA—principal coordinate analysis; PD—phylogenetic diversity; RCT—randomized controlled trial; RYGB—Roux-en-Y gastric bypass; SG—sleeve gastrectomy; TG—triglycerides; 2hPPG—2 h post prandial glucose.

**Table 3 medicina-57-01084-t003:** Alpha and beta diversity changes in included trials.

Alpha Diversity Indicator	Change	Changes in Beta Diversity	RCT	Metabolic Outcome Present
Shannon index	↑	Weighted, unweighted Unifrac: Dispersion	Cortez et al. [20]	↓ Anthropometric results
		No significant results in PCoA	Medina-Vera et al. [27]	↓ Glycemic, lipid profile, inflammatory results. ↓ FFAs
		No significant results in PCoA (Bray-Curtis distance)	Pedersen et al. [28]	↓ Glycemic results
	↓	–	Gu et al. (Acarbose arm) [17]	↓ Glycemic, lipid profile, inflammatory results
		Significant results in PcoA (↑: Bray-Curtis distance)	Zhang et al. (Symbiotic arm) [31]	↓ Glycemic, lipid profile results
		Significant results in PcoA (↑: Bray-Curtis distance)	Zhang et al. (Prebiotic arm) [31]	↓ Glycemic, lipid profile results
Simpson index	↑	Weighted, unweighted Unifrac: Clustering	Cortez et al. (CG) [20]	–
		Significant results in PCA, PCoA (↑: Bray-Curtis distance)	Tong et al. (Metformin arm) [18]	↓ Anthropometric, glycemic, lipid profile, inflammatory results
Inverse Simpson index	↑	No significant results in PCoA (Bray-Curtis distance)	Pedersen et al. [28]	↓ Glycemic results
Chao1 index	↑	Weighted, unweighted Unifrac: Dispersion	Cortez et al. [20]	↓ Anthropometric parameters
		Weighted, unweighted Unifrac: Clustering	Cortez et al. (CG) [20]	–
		No significant results in PCoA (UniFrac)	Lee et al. (RYGB arm) [22]	↓ Anthropometric, glycemic results
	↓	Significant results in: PCA, PcoA (↑: Bray-Curtis distance); ↑: weighted, unweighted UniFrac distances; abundance-weighted, ↑: binary Jaccard distances	Tong et al. (Prebiotic arm) [18]	↓ Anthropometric, glycemic, lipid profile, inflammatory results, ↑ dBP
		No significant results in PCoA (UniFrac)	Lee et al. (AGB arm) [22]	↓ Glycemic, anthropometric results
Species richness	↑	–	Murphy et al. (RYGB arm) [21]	↓ Anthropometric, glycemic results
		No significant results in PCoA (Bray-Curtis distance)	Pedersen et al. [28]	↓ Glycemic results
Observed species	↑	No significant results in PCoA (UniFrac)	Lee et al. (RYGB arm) [22]	↓ Anthropometric, glycemic results
	↓	No significant results in PCoA (UniFrac)	Lee et al. (AGB arm) [22]	↓ Glycemic, anthropometric results
Rarefaction	↓	–	Gu et al. (Acarbose arm) [17]	↓ Glycemic, lipid profile, inflammatory results
		Significant results in: PCA, PcoA (↑: Bray-Curtis distance); ↑: weighted, unweighted UniFrac distances; abundance-weighted, ↑: binary Jaccard distances	Tong et al. (Prebiotic arm) [18]	↓ Anthropometric, glycemic, lipid profile, inflammatory results
Phylogenic diversity	↑	No significant results in PCoA (UniFrac)	Lee et al. (RYGB arm) [22]	↓ Anthropometric, glycemic results
	↓	No significant results in PCoA (UniFrac)	Lee et al. (AGB arm) [22]	↓ Glycemic, anthropometric results
Gene count	↓	–	Gu et al. (Acarbosis arm) [17]	↓ Glycemic, lipid profile, inflammatory results
		Significant results in PCoA (↑: Bray-Curtis distance)	Zhang et al. (Symbiotic arm) [31]	↓ Glycemic, lipid profile results
		Significant results in PCoA (↑: Bray-Curtis distance)	Zhang et al. (Prebiotic arm) [31]	↓ Glycemic, lipid profile results
		No significant results in PCoA (Bray-Curtis distance)	Zhang et al. (Probiotic arm) [31]	↓ Lipid profile results

“–” means that a certain parameter was not evaluated, achieved, or provided in a specific trial. AGB—adjustable gastric binding; CG—control group; dBP—diastolic blood pressure; FFAs—free fatty acids; PCA—principal component analysis; PCoA—principal coordinate analysis; RCT—randomized controlled trials; RYGB—Roux-en-Y gastric bypass.

**Table 4 medicina-57-01084-t004:** Summary of significant phylum, genus, and species level microflora alterations after different type 2 diabetes treatment strategies.

More Abundant Phyla		*Firmicutes*				*Bacteroidetes*				*Actinobacteria*			
Intervention	Taxonomy	Phylum	Genus	Species	Overall	Phylum	Genus	Species	Overall	Phylum	Genus	Species	Overall
Surgery													
RYGB (3)		↑ (1/1)	↑ (2/2)	↑ (1/1)	↑	→ (1/2)	↑ (1/1)	(0/0)	?	↑ (2/2)	(0/0)	(0/0)	↑
AGB (1)		(0/0)	(0/0)	(0/0)	n/a	(0/0)	(0/0)	(0/0)	n/a	(0/0)	(0/0)	(0/0)	n/a
SG (1)		(0/0)	(0/0)	↑ (1/1)	→	↑ (1/1)	(0/0)	(0/0)	↑	(0/0)	(0/0)	(0/0)	n/a
Oral anti-diabetic medication													
Acarbose (2)		(0/0)	(0/0)	↑ (20/36)	→	(0/0)	(0/0)	↓ (14/16)	→	(0/0)	(0/0)	↑ (6/7)	→
Metformin (2)		→ (0/0)	↑ (2/2)	↑ (23/35)	↑	(0/0)	↓ (2/2)	↑ (5/5)	?	(0/0)	(0/0)	↑ (20/20)	→
Glipizide (1)		(0/0)	(0/0)	(0/0)	n/a	(0/0)	(0/0)	(0/0)	n/a	(0/0)	(0/0)	(0/0)	n/a
Probiotics (5)		→ (0/0)	↑ (3/3)	↑ (17/17)	↑	(0/0)	(0/0)	↑ (1/1)	→	(0/0)	↑ (2/2)	↑ (1/1)	↑
Prebiotics (6)		↓ (1/1)	↑ (8/12)	↓ (16/28)	?	(0/0)	↑ (4/7)	↑ (11/18)	↑	(0/0)	→ (1/2)	↓ (5/7)	?
Symbiotics (1)		(0/0)	(0/0)	↓ (24/41)	→	(0/0)	(0/0)	↑ (10/19)	→	(0/0)	(0/0)	↓ (5/6)	→
Less abundant phyla		*Proteobacteria*				*Verrucomicrobia*				*Fusobacteria*			
Surgery													
RYGB (3)		↑ (1/1)	(0/0)	(0/0)	↑	↑ (1/1)	↑ (2/2)	(0/0)	↑	(0/0)	(0/0)	(0/0)	n/a
AGB (1)		↑ (1/1)	(0/0)	(0/0)	↑	(0/0)	↑ (1/1)	(0/0)	?	(0/0)	(0/0)	(0/0)	n/a
SG (1)		(0/0)	(0/0)	(0/0)	n/a	(0/0)	(0/0)	(0/0)	n/a	(0/0)	(0/0)	(0/0)	n/a
Oral anti-diabetic medication													
Acarbose (2)		(0/0)	(0/0)	↑ (3/3)	→	(0/0)	(0/0)	(0/0)	n/a	(0/0)	(0/0)	↑ (1/1)	→
Metformin (2)		(0/0)	↑ (2/2)	↑ (18/22)	↑	(0/0)	(0/0)	(0/0)	n/a	(0/0)	↑ (1/1)	↓ (2/2)	?
Glipizide (1)		(0/0)	(0/0)	(0/0)	n/a	(0/0)	(0/0)	(0/0)	n/a	(0/0)	(0/0)	(0/0)	n/a
Probiotics (5)		(0/0)	(0/0)	(0/0)	n/a	(0/0)	(0/0)	(0/0)	n/a	(0/0)	(0/0)	(0/0)	n/a
Prebiotics (6)		(0/0)	↑ (1/1)	↑ (10/11)	↑	(0/0)	↑ (1/1)	↑ (1/1)	↑	(0/0)	(0/0)	↑ (1/1)	→
Symbiotics (1)		(0/0)	(0/0)	↑ (12/13)	→	(0/0)	(0/0)	(0/0)	n/a	(0/0)	(0/0)	↑ (1/1)	→

“↑” means that there is a tendency to increase within trials; “↓” means that there is a tendency to decrease within trials; “→” means that there is no tendency observed; “?” means that results are inconclusive; “n/a” means that no significant alterations were reported. Numbers in brackets provide the frequency of a predominant (increase/decrease) alteration/sum of all significant specific changes. E.g., ↑ (↓) (5/8) means that 5 out of 8 members from a phylum/genus/species level had increased (decreased) significantly. A specific tendency within a phylum, genus, or species was confirmed if the alteration ratio, where applicable, was greater than 0.5. If the ratio was equal to 0.5, it was observed as a case of no tendency. Overall tendency was confirmed (highlighted in green) if there was a unanimous alteration in at least two levels or it occurred in the phylum level. In cases of contradicting tendencies between levels, overall tendency was considered inconclusive (highlighted in yellow). AGB—adjustable gastric banding; RYGB—Roux-Y gastric bypass; SG—sleeve gastrectomy.

## Data Availability

All data can be found in the Appendix A online at https://www.mdpi.com/article/10.3390/medicina57101084/s1.

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
