# Peer review of "Microbiome Changes after Type 2 Diabetes Treatment: A Systematic Review"

_medicina, 2021, doi:10.3390/medicina57101084_

Round 1
Reviewer 1 Report
Your article evaluates the impact of different antidiabetic treataments on the intestinal microbiome and the associated improvement in different biomarkers. The articles is extensive and well realized, with adequate tables, appropiate metodology, great discussions and very detaileddiscussions of the included studies. However, the discussion can be improved. Please discuss if GLP-1 agonists impact gut microbiota in T2DM, you can consult Vesa CM. Diagnostics. 2020; 10(5):314. https://doi.org/10.3390/diagnostics10050314 and Corb Aron RA, Abid A, et.al, Microorganisms. 2021 Mar 17;9(3):618. doi: 10.3390/microorganisms9030618. PMID: 33802777; PMCID: PMC8002498.
Please discuss the impact of differnt drugs on the concentration of Akkermansia muciniphila if that was evaluated in your studies or otherwise discuss what you find in literature. Please discuss the impact of these drugs in reducing cardiovascular risk because of improvement of gut permeability and redcing inflammation, please check Babes, E.E.; Diagnostics 2021, 11, 850. https://doi.org/10.3390/diagnostics11050850 and Moisi MI, . Medicina. 2020; 56(3):118. https://doi.org/10.3390/medicina56030118
Author Response
We are grateful for the positive feedback and helpful comments for improving the review. Please see the attachment for more detailed response.

Reviewer 2 Report
MICROBIOME changes after type 2 diabetes treatment a systematic review
This is an interesting review. Literature on the microbiome is to me confusing and associations are interesting only when the importance of the association is proven. The introduction is well written and well referenced. The place of interventions to alter the microbiome and so to lower blood sugar and insulin resistance are referenced but the quality of these papers and whether the changes in the microbiome are the reason for diabetic improvement rather than an association is still to my mind uncertain . It is a pity that only systematic reviews are referenced
The aim of the study however still stands and the aim is only to evaluate microbiome and metabolic changes after different types of treatment in type 2 diabetes so the authors make it clear that their study is not evaluating
Changes in the microbiome as a cause of metabolic changes.
The material and methods section is concise.
The results section is more difficult. 16 suitable studies were found but they are very diverse as can be seen in Table 1.. 3.1 Oral anti-diabetic treatment Line 155’ alter microbiome and CONSEQUENT metabolic outcome. That suggests that the authors think changes in the microbiome Do change rather than are associated with metabolic changes.I would suggest that association be used unless the introduction is rewritten to convince the reader that microbiome changes do change metabolic outcome.
The description of the selected papers is through and the discussion well set out. The conclusions reasonable except the acceptance that there are beneficial bacteria( L611). I liked the comment ‘Where there are more questions than answers.
In conclusion a very difficult meta analysis well handled.
Author Response

(The authors gave the same response as above.)
